# Challenge of Utilization Vegetal Extracts as Natural Plant Protection Products

**Daniela Suteu [1,\*], Lacramioara Rusu [2], Carmen Zaharia [1,\*], Marinela Badeanu [3] and Gabriel Mihaita Daraban [1]**

[1] "Cristofor Simionescu" Faculty of Chemical Engineering and Environmental Protection, "Gheorghe Asachi" Technical University of Iasi, 73A D.Mangeron Blvd., 700050 Iasi, Romania; darabangabrielmihaita@yahoo.com

[2] Faculty of Engineering, "Vasile Alecsandri" University of Bacau, 157 Calea Mărăşeşti, 600115 Bacau, Romania; lacraistrati04@yahoo.com

[3] Faculty of Horticulture, "Ion Ionescu de la Brazi" University of Agricultural Sciences and Veterinary Medicine of Iasi, 3 Mihail Sadoveanu Street, 700490 Iasi, Romania; badeanumarinela@yahoo.com

\* Correspondence: danasuteu67@yahoo.com (D.S.); czah@tuiasi.ro (C.Z.); Tel.: +40-232-278683 (ext. 2260) (D.S.); +40-232-278683 (ext. 2175) (C.Z.)

**Abstract:** Natural plant protection products (known as biopesticides), derived from natural materials (plants, bacterial strains, and certain minerals) that can be used to control pests, are an alternative to plant protection chemicals (known as pesticides) due to certain advantages: less toxic to humans and the environment, no release/leaching of harmful residues, and usually much specific to the target pests. This review focuses on the systematization of information highlighting the main advantages related to the natural plant protection products used, the extractive methods of obtaining them, their physical-chemical analysis methodology, the specific constituents responsible for their pesticide effects, the mechanisms of action, and methods for direct application on vegetable crops or on seeds stored in warehouses, in order to eliminate the adverse effects occurred in the case of plant protection chemicals use. Special attention has been accorded to natural plant protection products from the spontaneous flora of Moldova (Romania's macroeconomic region NE), which can be considered a resource of valuable secondary metabolites, especially in the form of vegetable essential oils, with biological effects and biopesticide routes of action. All presented information concludes that biopesticides can successfully replace the chemical plant protection products on small farms and especially in silos (seeds and cereals).

**Keywords:** natural plant protection product (known as biopesticide); chemical composition; liquid-solid extraction; spontaneous flora from Romania; vegetal extract

## 1. Introduction

Ensuring food needs for a numerically growing population in certain geographic areas of the world has stimulated the use of natural plant protection products (as biopesticides) in agriculture because of their role in increasing the productivity of crops and protecting them from the action of certain pests [1,2].

In agriculture, the action of the pest manifests both during the development of the crop and in the post-harvest period (during transport, or storage). For this reason, the manufacturing of chemical plant protection products (known as pesticides) has been and is still closely related to certain branches of science such as biology, organic chemistry, biochemistry, as well as those of the agricultural sector.

The use of chemical plant protection products has led to increased productivity in agriculture mainly due its efficient action against the unwanted vegetation and removal of pests [2–4]. According to

data provided by the World Health Organization [1], chemical plant protection products usage was increasing worldwide since 1940 (when first use took place), especially in developed countries, having as main objectives the following [5]:

1. The control of arthropods during the vegetation period.
2. The control of pathogens from agricultural crops and in the forest.
3. Selective weed control in agricultural crops and forestry.
4. The disinsectisation/sanitation of silos, food stores, and greenhouses.
5. The treatment of seeds.

However, the intensive use of chemical plant protection products (persistent and non-biodegradable), often even irrational, has contributed to the initiation and intensification of deep soil degradation processes, groundwater and surface water pollution, and air contamination, consequently affecting the flora and fauna in the adjacent areas [2,6–11].

Contamination of crops from these soils has been recorded which has led to the impact on the entire food chain and especially on food products innocuity [12]. Therefore, new actual agricultural development directions are imposed and the development of new, highly efficient products that do not create dependence and resistance among the pests (thus reducing the number of applications and the dose of these products) to protect fauna and be environmentally friendly is urgently required [2,3].

In recent decades, international and national concerns have been intensified both to increase plant production and to improve its capitalization. Multiple and complex industrialization of plant crops is constantly evolving by applying modern methods for their preservation and processing.

The plant organism, in terms of the chemical composition and the functions that performs, is very complex. Today, plants are recognized as the sources of bioactive compounds (terpene, steroids, anthocyanins, anthraquinones, phenols, polyphenols, etc.) with applications in cosmetic, pharmaceutical, and food industries [3].

However, recent data show that a growing number of vegetal plants extracts have been studied and tested as natural plant protection products (biopesticides) (well-known are the extracts of *Chrysanthemum cinerariaefolium*, *Chrysanthemum coccineum*, *Haloxylon salicornicum*, *Stemona japonicum*, *Schoenocaulon officinale*, *Origanum vulgare*, *Thymus vulgaris*, *Azadiractina indica*, *Leuzea carthamoides*, *Mentha* spp., *Lavendula* spp., *Nicotiana* spp., and also certain active ingredients as polyphenols, alpha-chaconine, nicotine, thymol, silphinenes, carvacrol, pyrethrins, rotenone, byanodine sabadilla, from certain essential oils and plant extracts) that can be used against a wide range of arthropod pests that attack vegetal plant crops or seeds deposits. The obtained results demonstrated its high efficacy, multiple mechanisms of action and low toxicity on vertebrates [3,13–17]. However, the number of biopesticides based on vegetal plant extracts, which have been tested, is still low.

The aim of the work is to systematize information on the use of synthetic chemical plant protection products (pesticides) and their effects on the environmental factors and the quality of the agro-food products which motivates the advancement of the studies regarding the natural plant protection products (biopesticides) obtained from the vegetal extracts.

Ensuring the quality of people's life is closely associated to a series of concepts implemented in recent years, namely sustainable development, climate change prevention or sustainable and bio-dynamic agriculture, this research study presents some pertinent information on natural plant protection products which represent a real challenge for researchers and practitioners. Consequently, this work highlights the significance and impact of a few proved beneficial results of plant protection products in association with their specific characteristics, especially of the herbal extracts from the spontaneous flora of Moldova (Romania).

## 2. Started Point: Chemical Plant Protection Products

The synthetic chemical protection products are used in agriculture to prevent the growth and destruction of weeds and pests that affect the crop quality. Among the factors determining the alteration of

crops' quality and quantity are: wild animals, rodents, insects and, most of all, pathogenic microorganisms, such as: fungi (*Fusarium* spp., *Aspergillus* spp., *Lecanicillium lecanii*, *Beauveria bassiana*, *Trichoderma harzianum*, *T. harzianum*), bacteria (*Bacillus thuringiensis*, *Bacillus popilliae*, *Actinomycetes*, *Bacillus subtilis*, *Pseudomonas fluorescens*) and viruses (Baculoviruses, Microspridian—*Lymantria dispar*) [1–3,14,17,18].

The action of pests, of any category, is primarily to reduce the quality and quantity of crops and obtained food products, which may affect the health of consumer (e.g., human health) [7,10,13].

The specialized literature indicates the existence of several classification criteria of the chemical plant protection products [18]: origin, anti-harmful action (action on pests), physical form of presentation, chemical structure and/or degree of toxicity. In practice, the selection of certain chemical plant protection products is done in compliance with criteria for their actions on pests and toxic effects on the population and the environment [10,19], such as:

- Lower impact on human health and a very low toxicity to warm-blooded animals.
- Lower toxicity compared to other alternative pest control products.
- Lesser effects on other organisms in the environment, such as bees, fish, and birds.
- Lower potential for contamination of soil and, implicitly, groundwater.
- Minimum number of required applications.
- Higher efficiency at lower doses and avoidance to pest resistance.

Last but not least, the selection of a particular chemical plant protection product is made according to the purpose and its main characteristics: selectivity and specificity, systemic action, phytotoxicity, occurrence of pest resistance, toxicity to humans and animals, environmental pollution, etc. [1,6,7,10,19].

Practical use of chemical plant protection products involves direct application to soil/plants, in relatively low doses, in the liquid form or mixtures. Particular attention is paid to the modulation of these compounds, which involves mixing the active substance with various solid ingredients (talc, kaolin, clays, calcium carbonate, etc.) or liquids (water, organic solvents, mineral and/or essential oils), because they must allow uniform spread over large areas of reduced amounts of active substance and achieve an improvement in adhesion to plants and thus ensure an optimal biological effect [19].

Moreover, the choice of conditioning form and formula helps to increase the effectiveness and economics of chemical plant protection product treatments and reduces the risk of pollution and phytotoxicity. Of great importance in the chemical plant protection product conditioning is also the choice of auxiliary products (solvents, surfactants) which should be based on a number of factors dependent on the intended purpose and the final properties of the commercial product [19]: the physicochemical properties and the active substance action; the physiological and biological characteristics of the pest; the type of culture being treated; the soil composition; the climatic conditions, etc.

The chemical plant protection products, under different trade names that depend on the manufacturer, are conditioned in various forms to fulfill much user requirements:

(a) Solid products such as: powdered powders, wettable powders, granules, impregnated strips, solid fertilizer mixtures, etc.
(b) Liquid products, namely: aqueous solutions, mixtures of organic solvents or mineral oils, which can provide a fine spraying and, in the aerosols, emulsifiable concentrates.
(c) Micro-encapsulation and fixation on macromolecular support is a modern conditioning process, with advantages over classical methods, applied in the cases where the structure and properties of active derivatives allow such approach.

A good chemical plant protection product should be strong against pests, should not endanger the human health and non-target organisms, and ultimately should break down into harmless compounds and thus it does not persist in the environment. Both relative and specific toxicity of the chemical plant protection products should be estimated to determine its potential and it is closely related to the spray droplet size and chemical dose density, the application time, which can also provide adequate control



of the pests. Also, the research on the development of adequate packaging and disposal procedures is needed, as well as application of equipment optimization. All these will lead to the rational use of chemical plant protection products, and thus they can be used in an acceptable and sustainable way.

Very strict laws (i.e., Environmental Protection Agengy—EPA's regulation of pesticides (EPA Agrees to Regulate Novel Nanotechnology Pesticides after Legal Challenge-March 2015); (EC) 1107/2009) should be adopted to protect the flora and fauna surrounding the environment in which the target pests or target organisms live.

A number of indications/recommendations on the chemical plant protection product label can prevent the effects on non-target organisms. However, there are many cases where the benefits of chemical plant protection products (pesticides) come with a number of disadvantages due to their toxicity or their degradation products, to treated crops or to the flora and fauna adjacent to these crops [10].

## 3. Next Level: Natural Plant Protection Products (Biopesticides)

Today, the concern for people's life quality has become more obvious by ensuring the protection of the environment and food safety. Food quality is closely related to raw material production conditions. Vegetable crops, seed production and preservation are influenced by the quality of pest control products and management of weeds as well as parasitic crops. A few regulations were permitted to reduce the usage of synthetic chemical plant protection products in agriculture where it was possible [20].

At the same time, it was expanded the phenomenon of resistance of the insects and harmful organisms to the action of synthetic chemical plant protection products dispersed on vegetal plants as a result of repeated use and increased doses [21]. In this context, new concepts appear and develop, such as the nano-biopesticides [22–24] and ecological organic agriculture. The concept of ecological organic agriculture does not accept the use of chemical synthesis compounds in order to protect, stimulate the growth or preservation of plants, fruits, and seeds [1,25], and thus a new category of compounds, natural plant protection products, appeared as a challenging alternative.

Natural plant protection products (biopesticides) are natural compounds obtained from the mineral, plant, or animal kingdom, including microorganisms, by various methods of extraction of active compounds or classes of active compounds/ingredients with beneficial effects in pest control through various mechanisms [13,20,26–28].

The natural plant protection products are considered the third-generation of chemical plant protection products (pesticides) that are rapidly gaining popularity. Starting with the 19th century, agriculture begun to use natural insecticides such as pyrethrum and rotenone, derived from chrysanthemum flower powders and *Derris* roots [20]. A wide range of microbial products derived from microorganisms and other natural sources have been added to its [13,15,21,29,30]. The natural compounds of the general category of biopesticides may belong to the next subclasses: biofungicides (e.g., *Trichoderma* spp.), bioherbicides (*Phytophthora* spp.), bioinsecticides (spores forming bacteria, *Bacillus thuringiensis* and *B. popilliae*, *Actinomycetes*), entomopathogenic fungus (*Beauveria bassiana*), microscopic roundworms (Entomopathogenic nematodes) [31], spinosad (*Saccharopolyspora spinosa*), insect hormones, and insect growth regulators [3,14,17,29].

Among the natural plant protection products, bioinsecticides used in the protection of crops and seeds in storage conditions and the management of weed control are of particular interest.

There are known situations when it was applied different volatile oils obtained from different plants for their bioinsecticide effect, although the mechanism of action is insufficiently known [16,21,31–33]. In a recent review, Zoubiri and Baaliouamer (2014) make a remarkable systematization for an impressive number of plants (227) belonging of many botanical species (259) from which volatile oils with a very complex chemical composition were obtained (i.e., spathulenol (8.6%), caryophyllene oxide (7.5–10.6%), cis-3-hexen-1-ol (11.3%), 1-hexanol (5.8%), β-caryophyllene (9.7%), γ-cadinene (5.0%)), which showed insecticidal activity [16].

The classes of biologically active compounds in most essential vegetable oils are terpenes and terpenoids (~25,000 types, 55%), alkaloids (~12,000 types, 27%), and phenolic compounds (~8000 types, 18%) which can be framed into two major classes: phytoalexins which are compounds that are synthesized *de novo* (as opposed to be released by, for example, hydrolytic activity) and phytoanticipins which are pre-formed infectious inhibitors [32]. The authors also systematized the results of some studies performed for determination of the essential vegetal oils' effect on insects. For this purpose, for the type of tested insect, the applied dose was expressed as: LD (lethal dose) 50/90/95; LC (lethal concentration) 50/80/90/100; RD (repellent dose) 50; ED (effective dose) 50; LT (lethal time); MLD (minimum lethal dose); RC (repellency class), respectively the insecticidal activity type (larvicidal, contact, fumigant, repetitive, acaricidal, antifungal, adulticide) [16].

Toxicity of certain active ingredients of some natural plant protection products extracted from plants against insects varies greatly depending on type of solvent used as carrier, such as shown in Table 1 for the oral and dermal lethal dose ($LD_{50}$) [15] or for fumigant toxicity of certain plant essential oils (EOs) (e.g., for *Callosobruchus chinensis* EO − $LC_{50}$ = 10.8–11.0 µL/L after 24 h (LT), or *Senegalia Tenuifolia* EO with a concentration of 12.5 µg/mL air (MLC) caused 96.6% mortality in larvae of *Lycoriella Ingenue*) [34].

**Table 1.** The oral and dermal ($LD_{50}$) lethal dose of some ingredients/active compounds.

| Ingredient | Oral $LD_{50}$, [mg/kg] | Dermal $LD_{50}$, [mg/kg] | Ingredient | Oral $LD_{50}$, [mg/kg] | Dermal $LD_{50}$, [mg/kg] |
|---|---|---|---|---|---|
| Nicotine | 50–60 | 50 | Linalool | 2440–3180 | 3578–8374 |
| Rotenone | 60–1500 | 940–3000 | Neem oil | >5000 | >2000 |
| Pyrethrins | 1200–1500 | >5000 | Pongam oil | >4000 | >2000 |
| D-limonene | >4000 | >5000 | Ryania | 750–1200 | 4000 |

Another relevant example is the study of Bett and his collaborators (Miller and Smith), which systematizes a series of results of recent research on the repellent effect and contact toxicity of essential oils of *Cupressus lusitanica* (Miller) and *Eucalyptus saliga* (Smith) on insects that attack agro-food products (seeds, grains) during storage [14].

Recently, the development of new classes of natural plant protection products has been made and succeeded, being approved and integrated into agricultural practice with the support of institutions empowered to market of these products with efficacy on pests, which has been a real success for commerce [35]. Among these, good examples include:

1. Rosemary essential oil (commercial name: Ecotrol™, Sporan™, Ecosmart) used as insecticide, acaricide, fungicide (octopamine antagonists; membrane disruptors) [1,15,20,21,32].
2. Clove essential oil (commercial name: Matran EC, Burnout II, Bioorganic Lawn) used as insecticides, herbicides (neurotoxic, interference with the octopamine neuromodulator) [32,36].
3. Jojoba essential oil (commercial name: Detur, E-Rasem, Eco E-Rase, Permatrol, Erase™) used as fungicide, insecticide (β-suffocation) (eggs and immature life stages), repellent, blocking access to oxygen) [1,21,32].
4. Cinnamon essential oil (commercial name: Weed Zap™, Repellex, Ecosmart) used as an insecticide, herbicide (octopamine antagonists, membranes disruptors) [15,20,21,32,36].
5. Lemon grass essential oil (commercial designation: GreenMatch EXTM, EcoPCO) used as insecticides, herbicides (octopamine antagonists, membrane disruptors, etc.) [1,15,20,21,32].

Although these alternative natural plant protection products are natural and have repellent, vermifugal, germicid, and vermicid effects, the widespread practical applicability is still limited due to farmers' reluctance to accept natural products as biopesticides and due to the insufficient studies in this area [13,25,32,37,38].

Particular attention is paid to natural compounds with bioinsecticidal action that can be obtained from a variety of plant, animal, or microorganism sources [15,20,30]. The antibacterial action manner

depends on the characteristics of the insecticide classes to be destroyed by shock effects, ovicidal or larvicidal effects, multiplication influence (pheromones). Even though the number of plant-originated insecticides of practical importance is rather limited compared to chemical plant protection products, the continuation of finding new active compounds/ingredients of different plants in the entire world is the future challenge in this field.

Insecticides of plant origin are largely nontoxic to humans and animals, do not leave toxic residues and do not cause the occurrence of insect resistance [3,14–16,20,21,39].

The relatively high price of natural insecticides limits their application to restrictive areas, such as sanitation, greenhouses, etc. Among the compounds with biological action against insects, obtained from the plant species, there are remarkable various categories of alkaloids, pyrethrins, rotenoids and terpenic structure-based insecticides [15,20,30]. The mechanism of insecticide action is shown in Figure 1.

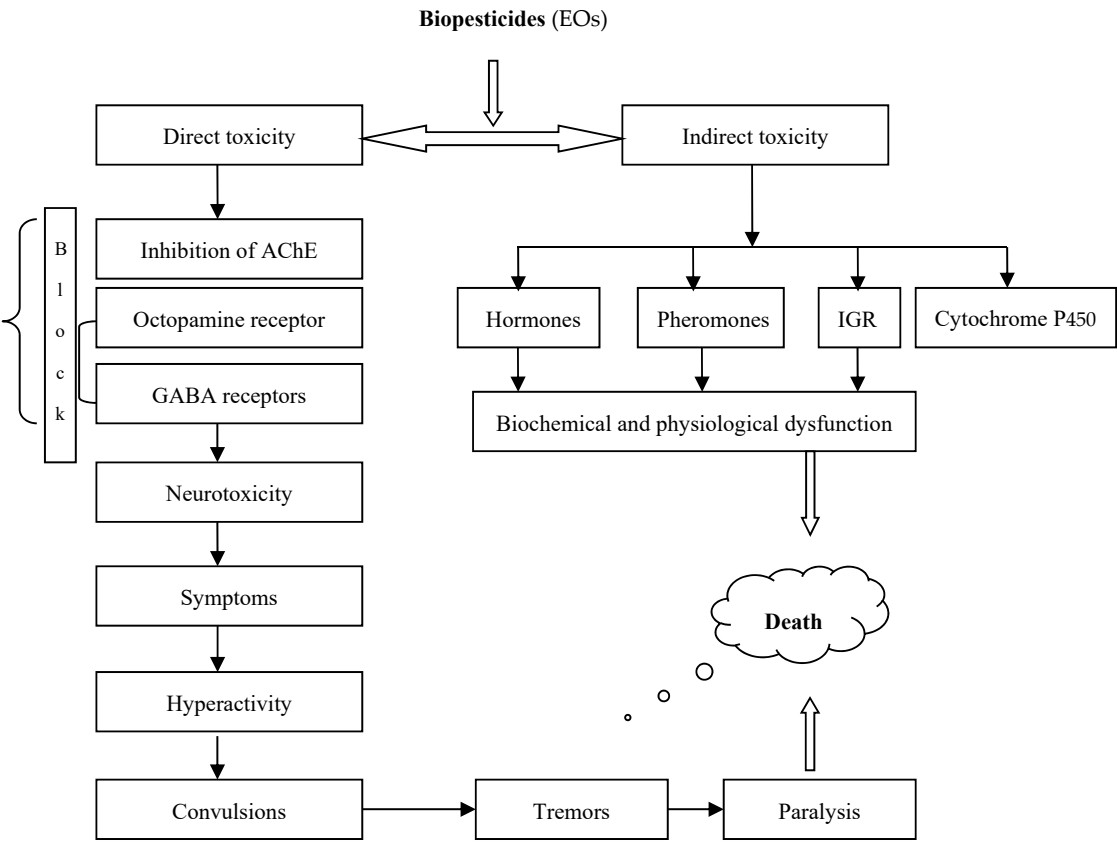

**Figure 1.** The action mechanism of essential oils on insects (adapted from Mossa 2016 [21]). (*Notes*: *EOs*—essential oils; *AChE*—acetylecholinestrase; *GABA* receptor—γ-aminobutyric acid receptor; *octopamine*—a neurotransmitter, neurohormone and circulating neurohormone-neuromodulator working on nervous system in insects).

## 4. Plant Extracts as Natural Plant Protection Products

Plants can be considered as a perfect laboratory with potential to supply organic and inorganic substances without which human and animal life would be impossible [15,27,32,40,41]. These substances synthesized in the plant kingdom may be of primary (proteins, carbohydrates, and fats) or secondary (terpene, steroids, anthocyanins, anthraquinones, phenols and polyphenols, etc.) metabolites classes [3,16,32,39,42–44].

Secondary metabolites reported till now include the following classes: nicotine, caffeine, flavorings (vanillin and cinnamic aldehyde), pigments (indigo and kaki), pyrethrum (natural insecticide obtained from marigolds and chrysanthemum varieties), rubber (natural latex obtained from the bark

of the rubber tree), dyes, odorants, flavorants and alkaloids [3,16,39,42,45]. Although essential oils have demonstrated high efficacy, multiple mechanisms of action, low toxicity on vertebrates, and potential use of byproducts as reducing agents, the number of commercially available natural plant protection products based on essential oils still remains low.

Practical use of vegetal extract as natural plant protection products is limited to individual producers and small crops. Herbal extracts used as biopesticides have a few advantages such as:

- High efficacy against a wide range of pests and diseases of agricultural and medical importance.
- Multiple action mechanisms due to the large number of active ingredients in each mix.
- Low toxicity against non-target organisms, including humans.
- Relatively simple and cheap production processes.
- Reducing health risks during application due to low residue toxicity.

Analyzing the main strengths and weaknesses that result from the use of plant extracts as natural plant protection products, the main challenges for future research include: (1) developing effective stabilization processes (e.g., microencapsulation); (2) simplifying complex and costly authorization requirements for natural plant protection products use, and (3) optimizing plant growth conditions and extraction processes leading to a homogeneous chemical composition [39].

*4.1. Methods of Vegetal Extracts Preparation*

The quality of products obtained by plant processing (which spontaneously grows and develops spontaneously or is grown under controlled conditions, such as medicinal, aromatic, ornamental or wild) is given by the content of active principles. This content in turn is dependent on ecological factors, the species growth area, the culture technology, the biological value of the cultivar (population, variety, hybrid, etc.) and, last but not the least, the processing modalities.

Obtaining active principles with different properties and applications (in making products for phytosanitation and phytopharmaceutical uses, pharmaceuticals, foodstuffs, or cosmetics) from vegetal material can be achieved mainly by extraction methods. They were especially noticeable solid-liquid extraction working with different fragments of the plant or in plant mixtures dependent on established target [42,44–52], and liquid-liquid extraction for separation of certain components from the crude extract of the solid-liquid extract.

The studied literature indicates many variants of classical or modern solid-liquid extraction that can be successfully applied for this purpose. Thus, Talmaciu et al. [42] systematize in their work some variants of liquid-solid extraction presented in scientific literature [43,49,51–53]. It started from the classical ones (solvent extraction and distillation) to modern variants or unconventional ones (pulsator—electric field, enzyme digestion, ultrasound, microwave heating, or supercritical fluid extraction) or more recent date methods entitled "green extraction techniques", which use modern solvents to extract the active principles of the plant.

The choice of an extraction technique depends on the purpose of the application, the type and the part of the plant to be used, the solvent requirements (acceptable level of toxicity on the extract and the environment, chemical nature of the compounds to be extracted), temperature regime, and optimum time without initiating the destruction of chemical compounds (Table 2), and follows the methodology described in Figure 2.

Processing plants for their use as extraction material of bioactive compounds involves two process phases:

(1) Primary processing consisting of the drying, conditioning, and packaging of plants.
(2) Advanced processing, which involves the transformation of raw materials obtained from primary processing into products that can be marketed: phytotherapeutic products (aqueous extractive solutions, hydroalcoholic extractive solutions, lyophilized powders from extractive solutions), phytosanitation products, cosmetics, nutritional supplements and dietary supplements or food additives.

Table 2. Methods used to obtain plant extracts.

| Plant/Common Name | Extraction/Analysis Method | Extract Composition/Pharmacological and Phytosanitary Effects | Ref. |
|---|---|---|---|
| *Achillea millefolium* L. commonly known—as Yarrow | EOs extraction (0.4–1.1%) was carried out by hydrodistillation of dried plant samples (15–20 g; leaves, flowers and rods) for 2.5 h/ Extract analysis was carried out by GC/MS method (e.g., HP 5890 chromatograph/HP 5971 Mass spectrometer (70 eV ionization), equipped with capillary column (50 m × 0.32 mm, film thickness 0.25 μm). The heater temperature was varied between 70 °C and 250 °C with specific controlled increasing rates and maintained at 250 °C using He as carrier gas (2.0 mL/min). Alcoholic extraction studies (dry plant—ethanol) were also performed (more polar constituents obtained than in aqueous extracts). | *Major representative constituents* (isolated as ingredients from plant EOs): chamazulene, sabinene, β-pinene, 1,8-cineles, linalool, α-thujone, β-thujone, ocimene, camphor, ascaridole, caryophyllene oxide, β-eudesmol and α-bisabolol/ *Pharmacological and phytosanitary effects:* antimicrobial action as antiseptic, analgesic, anti-inflammatory and wound healing. | [54,55] |
| *Origanum vulgare* L., commonly known as—Wild Marjoram | Solid (aerial part of dry plant)—liquid (solvent) extraction with separation of EOs and aqueous, alcoholic (methyl and ethyl alcohol) and organic extracts (in cyclohexane, hexane, dichloromethane, ethyl acetate)/ Extract composition was controlled by GC/MS analysis of representative constituents for each extract or EO. Different conditions such as low pH, temperature or oxygen levels enhanced the antibacterial action of essential oils [4,9–11,15]. | *Major representative constituents*: phenolic compounds such as carvacrol and thymol, γ-terpine [10], linalool, p-cymene, 4-terpinol, β-caryophyllene, other terpenoids and flavonoids, as well as various organic acids such as rosmarinic acid, oleanolic and ursolic acid [20,24,25,30]/ *Pharmacological and phytosanitary effects:* antibacterial, antifungal, antioxidant, anti-inflammatory, antitumor and antiviral. There are used as infusions [34,40] | [56] |
| *Satureja hortensis* L., common known as—Thyme | EO extraction was done by hydrodistillation using a Clevenger type device (100 g dry powder plant, 3 h-extraction; water separation from EO by filtration over an anhydrous sodium sulphate layer)/ EOs and aqueous extracts were analyzed by gas chromatography (e.g., HP 6890 chromatograph with HP1/SPB-1/SupelcoWax-10 silicon-based column, thickness 0.25 mm) coupled with mass spectroscopy (e.g., HP 5973 selective mass detector, Agilent technologies). | *Major representative constituents*: —in *EOs*: thymol (45.9%), gama-terpinenes (16.71%), carvarol (12.81%) and p-cymene (9.61%); —in *aqueous extracts*: thymol (63.40%), linalool (42%), α-pinene (27.87%), b-pinene (22.10%), zingiberene (31.79%)/or linalool (42%), thymol (25.10%) and α-terpineol (10%)/or α-pinene (27.87%), 1,8-cineole (20.15%) and linalool (10.26%) | [57] |
| *Calendula officinalis*, commonly known as—Marigold | Extraction with a solvent (acetonitrile and ethyl acetate or 1% glacial acetic acid) using only the flowering plant, dried, crushed and sieved (<2 mm); extract separation by centrifugation/ Separated extracts were analyzed by liquid chromatography (LC Agilent 1200 column XDB-C18/4000 QTRAP LC-196) coupled with MS) (e.g., LC-MS system) (SANTE guidelines) | *Representative constituents* for antimicrobial activity in extracts: high content of total polyphenols, flavonoids and tannins as well as various synthetic pesticide residues. | [58] |
| *Pimpinella anisum*, commonly known as—Anise | EO extraction by hydrodistilation using only the dried and crushed seeds; water separation by filtration over an anhydrous sodium sulfate layer (after stored at 4 °C)/ *P. anisum* extract and EO were analyzed using UPLC MS Waters Alliance system/USA, with self-collecting and diode waters 2996 | *Major representative constituents*: —in EOs (LC-MS analysis): phenyl-propenoids, mono- terpene such as *trans*-anethol (82.1%), *cis*-anethol (5.8%), estragol (metilcavicol) (2.5%), linalool (2.3%), α-terpineol (1.5%) and metil-eugenol (1.3%) | [59] |
| *Urtica dioica*, commonly known as—Nettle | EO extraction with alcoholic solution (ethanol and/or methanol)—water (1 g/10 mL alcohol and distilled water (8:2 v/v) using dried leaves (fraction < 14 in); phases separation by centrifugation (3000 rpm, 15 min)/ EOs were analyzed by liquid chromatography coupled with mass spectroscopy (LC-MS) | *Major representative constituents*: high content of hydroxy-cinnamic acids (chlorogenic acid, caffeic acid, rosmarinic acid) and flavonoids (quercetin) | [60] |

**Table 2.** *Cont.*

| Plant/Common Name | Extraction/Analysis Method | Extract Composition/Pharmacological and Phytosanitary Effects | Ref. |
|---|---|---|---|
| *Hypericum perforatum* | Alcoholic extraction (80% ethanol solution) by by reflux distillation (8 h, 60 °C water bath) using only dried leaves (60-mesh); separation of soluble/insoluble extracts by centrifugation (4000 rpm)/filtration/drying/ Extracts/EOs were analyzed by GC-MS system (e.g., Agilent GC-MS system equipped HP-5MS column (30 m × 0.25 m), t = 150–300 °C, 6 °C/min | *Main representative constituents:* polysaccharides (carbohydrates) and monosaccharides, L-arabinose (40.68%), D-galactose (36.72%), manose (3.58%), ramnose (14.38%), glucose (2.54%), xilose (1.83%), piranose, uronic acid, etc. | [61] |

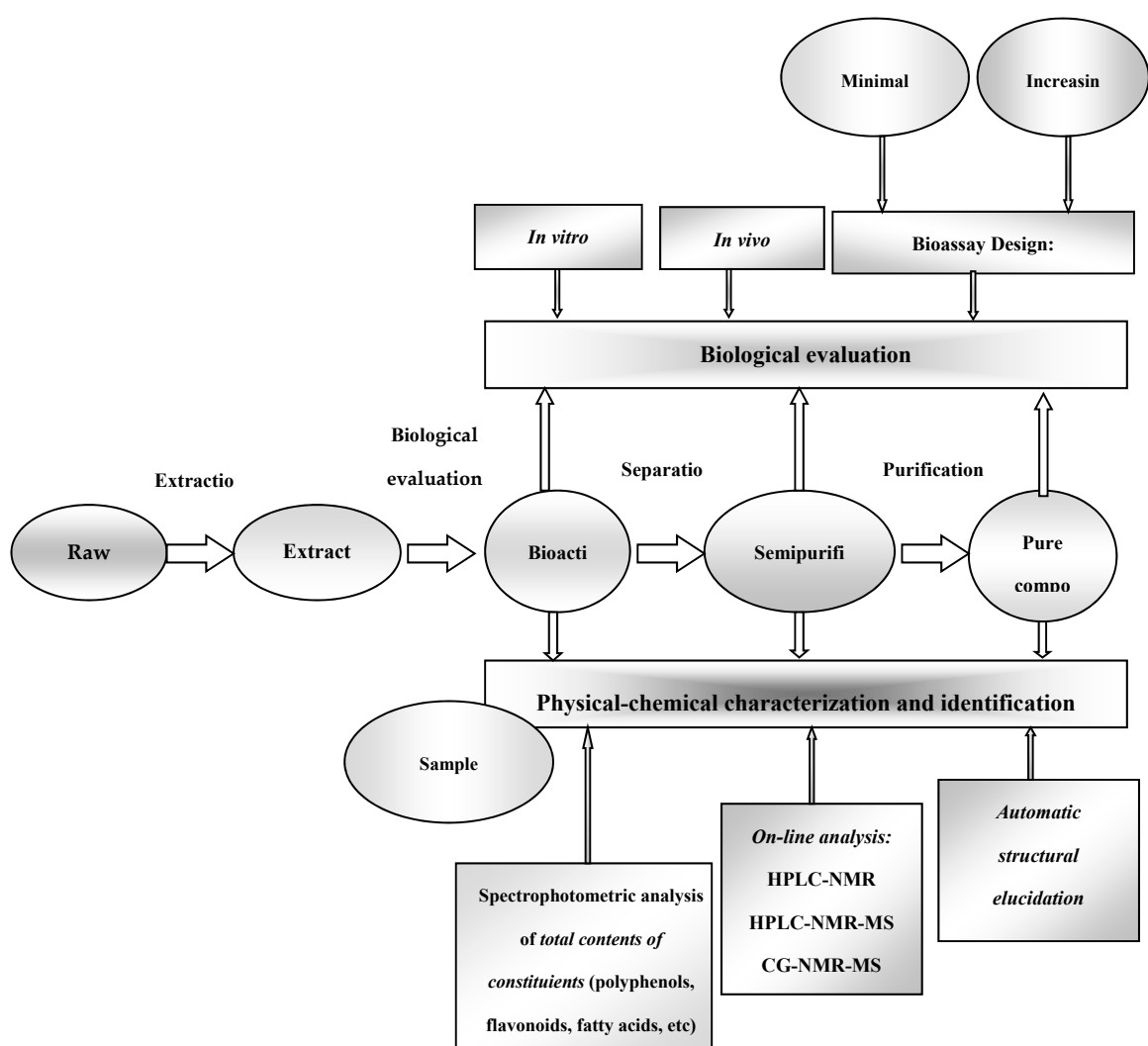

**Figure 2.** Schematic representation of the extraction of active principles from different plants [32].

Because these active principles with biological action on living organisms can be exploited, they must be conditioned in various forms accessible to the consumer but also effective in releasing the active principle in certain types of media (aqueous, alcoholic, etheric or other organic types) (Table 3). Vegetal products may be used in the following forms: (i) natural state: whole, fragments, powders or as influenza/infusion, decoction products, macerate, or (ii) phyto-sanitary preparations (extracts, tinctures, syrups) which are easier to administer and not very expensive.

Solvent extraction has been shown to be the most commonly used bioactive compound extraction method/technique for plants, involving the extraction of components from a solid or semi-solid sample

in a suitable solvent [62–66]. It can be practiced in several variants depending on the type of compound to be obtained and the cost-benefit criterion: (a) the classical variant, (b) the supercritical fluid variant, (c) with natural deep eutectic solvents or as vegetal oil, and (d) with alternative heating source (ultrasounds and microwaves) [44,46,47,52,53,67]. In the extraction technique, the choice of solvent (water, organic solvent (ethanol, methanol), or mixtures of organic solvents (ethyl acetate, hexane, etc.) is essential and depends on the nature of the substance to be extracted and the subsequent treatment to be applied further to the obtained extract [46,48,52,53].

The development of extraction techniques to obtain highly effective separation of all-natural ingredients, irrespective of their concentrations, has prompted numerous comparative studies on the advantages and disadvantages of these separation techniques currently practiced [42,45,63]. Numerous restrictions on organic solvents, linked to toxicity and environmental impact, have boosted the identification of new extraction opportunities with modern or "green" solvents.

Green technologies (e.g., sonoextraction, microwave extraction) have emerged and developed [42,47,49,51,67,68] due to a few appreciable advantages: (a) short extraction time, (b) low energy consumption, (c) use of non-toxic environmental solvents (environmentally friendly ones). Also, another direction in solvent extraction is the use of non-polluting "green solvents" that are non-volatile organic compounds with high dissolution, low toxicity, low environmental impact, which are usually obtained from renewable resources [46,48,49,52].

### 4.2. Methods for Determination of the Chemical Composition of Plant Extracts

The chemical composition of plant extracts depends mainly on the parts of the plant that are subjected to extraction, the growing and harvesting conditions, as well as the extraction technique and conditions. The resulting extract, irrespective of the extraction technique used, has an extremely complex composition, which is why the selectivity and sensitivity of the quantitative determination method are essential. The plants are distinguished by their qualitative chemical composition, largely on compound classes, but different as a quantitative composition, as well as by the presence of compounds that differentiate them by their taste, smell, and biological properties. In general, a number of classes of main organic chemical compounds (carbohydrates, flavonoids, phenols, quinones, saponins, tannins, lipids, fatty acids) are distinguished in different proportions along with bitter principles, resins, anthocyanins, gums, dyes, microelements (Ca, Mg, K, Zn, Cu), and vitamins [17,21,33]. The volatile oils obtained have different colors and consistencies from one plant to another and are characterized by a complex content rich in polyphenols, oxygenated monoterpene compounds (such as thymol, carvacrol, linalool, borneol, carvone, camphor, etc.), hydrocarbons (p-cymene, α- and β-pinene, myrcene, limonene, etc.) (Table 2), and sesquiterpenes [39,69].

Considering the selectivity and sensitivity as criteria for assessing the analytical performances for a method of quantitative determination of the composition of plants, extracts and oils obtained from them, a series of methods applied for quantitative or structural determinations are practically preferred depending on the context. Frequently used quantitative determination methods include the following: spectrophotometric methods (SM), high performance liquid chromatography (HPLC), high speed counter current chromatography (HSCCC), and thin-layer chromatographic (TLC) [45,65,69–71]. For structural characterization, the scientific literature mentions spectrometric methods as well as Nuclear Magnetic Resonance (NMR), Fourier Transform Infrared Spectroscopy (FTIR), Mass Spectroscopy (MS) [42,44,45,69–71].

Achieving high performance in terms of efficiency, costs and time of analysis was possible by applying combined separation—determination techniques, such as: gas chromatography—mass spectroscopy (GC-MS); liquid chromatography—mass spectrometry (LC-MS) and flow injection electrospray ionization—MS (ESI-MS) (Table 4) [65,72–79].



**Table 3.** Methods for extraction of active compounds from plants [42,44,46].

| Extraction Method | Common Solvents Used (Volume) | Dried Plant Amount/Extraction Time | Characteristics/Advantages/Disadvantages |
|---|---|---|---|
| With solvents | Methanol, ethanol, or mixture of alcohols and water (repetable volume of 50–200 mL; temperature dependent of solvent used) | (5–5000) g/3–4 days | - Numerous methods to extract ingredients of interest (initially as a mixture of active and inactive components, further fractionated to allow their individual identification). <br> - It is expensive due to the need to dispose off large amounts of organic waste (waste solvent) with potential environmental risk or often to incinerate its (environmentally hazardous and costly process). |
| Distillation | Water—steam system (*hydro-distillation*), i.e., <br> (i) water distillation—for dried plant material; <br> (ii) water and steam distillation—for both fresh and dried plant materials; <br> (iii) direct steam distillation—for fresh matter | (5–100) g/3–12 h | - It is often simple and practical, but selective for only a few types of compounds, although it depends on solvent used. <br> - Steam distillation is considered as traditional method, most common and economical one. <br> - Unfortunately, it is tedious to operate, employing hazardous solvents, requiring additional steps to remove, and many employing heat, thereby resulting in the degradation of heat-labile molecules. |
| Maceration | Methanol/ethanol, or mixture of alcohols: water, 1:10 (not too much applicable; room temperature) | (0.5–100) g/3–7 days (samples imbibated with solvent processed on rotary evaporator (50 °C, 100 rpm, 120 min) | - It is obtained a mixture of active and inactive components which are further fractionated to allow their individual molecular identification. <br> - It is expensive due to the need to dispose of large amounts of organic waste (risk control). <br> - It demands filtration as finishing extraction step. |
| Mechanical extraction | - No/limited solvent use; <br> - It is dependent on equipment used. | (5–5000) g/10 min–1–2 h | - Is used mainly for extraction of oil and juice; <br> - It does not require external heat and solvent limited application; <br> - Is a non-selective method. |
| Soxhlet extraction method | Methanol/ethanol, or mixture of alcohols and water (150–200 mL; low heating support), other solvents | (10–100) g/(3–18) h | - Soxhlet distillation is considered as traditional method used for several decades; traditionally used for the extraction of lipids. <br> - It is expensive (need to dispose of large amounts of organic waste) and is tedious to operate, and often employs hazardous solvents causing degradation of heat-labile molecules. |
| Enzyme digestion | - mixture of enzymes (cellulase, pectinase and protease, ratio 2:1:1) <br> - a combination of cellulase with a carbohydrase multi-enzyme complex | (10–100) g/(3–10) h (70 °C, 450 bar/510 min in pre-treatment of pomegranate peel, or 50–60 °C, 170–260 bar/20.64 min in extraction of lycopene from tomato skin) | - Enzyme digestion marked an increase in retrieval of bioactives compared to other conventional extractions (even increase the performance of a single step of supercritical fluid extraction (SFE) when it is used in combination); <br> - It depends significantly on the performance of enzymatic digestion highly influenced by operating regime (temperature <80 °C, pressure < 450 bar, operating time < 10 h) and metabolism activity. |

Table 3. *Cont.*

| Extraction Method | Common Solvents Used (Volume) | Dried Plant Amount/Extraction Time | Characteristics/Advantages/Disadvantages |
|---|---|---|---|
| Ultrasound heating (sonication) | Methanol/ethanol, or mixture of alcohol and water (50–100 mL) associated with ultrasonic-assisted installation | (5–100) g/(0.15–1) h (e.g., olive leaves) Solid-liquid oil enrichment (10:1, 20 min/25 °C) with ultrasound (225 W, 50% amplitude, duty cycle 0.5 s) produced edible oils with better quality than nonultra-sonicated oils. | - Rapid extraction; relatively low additional cost; small amount of solvent; <br> - Non-selective method; faster energy transfer and response to process control; <br> - Moderate extraction time (10–60 min); reduces thermal gradients and improves the extraction yield; <br> - It requires additional steps (filtration); <br> - Example: oil enrichment with phenolic compounds by ultrasonic maceration of leaves (60 W, 16 °C and 45 min), e.g., oleuropein from olive leaves—total phenolic content of 111.0–414.3 mg of oleuropein equiv./kg of oil, and alpha-tocopherol of 55.0 g/kg of oil. |
| Microwave heating (with traditional extraction) | - Small amounts of solvent (methanol/ethanol, or mixture of alcohol and water); <br> - Microwaves are nonionizing electromagnetic waves (frequencies in range of 300 MHz-300 GHz), positioned between X–IR rays | (5–100) g/(0.15–1) h (e.g., olive leaves) | - Rapid extraction; small amount of solvent; relatively low additional costs; non-selective (large number of compounds extracted); <br> - Use of high pressure and temperature; <br> - Limited amount of sample; fully automatic operation; <br> - Heating with microwave energy is acting directly on the molecules by ionic conduction and dipole rotation, and only selective and targeted materials can be heated based on their dielectric constant. |
| Supercritical fluid extraction (SCFE-based method) | - Supercritical fluids used as solvents (50 to 300 mL), i.e., propane, n-butane, methyl-propane, freon, nitrous oxide, dimethyl ether (methoxymethane), or water (critical temperature is almost 400 °C) <br> - Carbon dioxide ($CO_2$) in its supercritical fluid state (sc$CO_2$) is the most commonly used SCF solvent being inert, recyclable, non-flammable and nontoxic. | milli-grams (<0.008 g) to grams (<100 g)/few minutes-often within 1 h (SCF extraction: batch, continuous flow or semi-con-tinuous, 200–1000 W) $CO_2$ has a low critical t = 31.1 °C) and relatively low critical pressure (72.8 bar) | - It requires relatively short processing times (rapid extraction), small amount of organic solvent or no solvent; reduced processing energy inputs/an alternative solvent approach. <br> - It produces extracts with little or no organic co-solvent, or no solvent residue, being suitable for thermo-sensitive compounds, production of cleaner extracts; selective extraction (small number of extracted compounds. <br> - It is able to extract bioactive molecules minimising degradation and preserving thermally labile compounds; facilitate collection of pure $CO_2$ solvent (gas); usually inexpensive to operate/run. <br> - Disadvantages: the high establishment cost, and the selective solvent nature of $CO_2$. |
| "Green extraction techniques" (with, green solvents') | - Non-polar and lipophilic systems with highly variable and complex composition, depending on their origin, quality and producing methods; <br> - Major component is glycerine. | (5–1000) g/(2–18) h | - Vegetal oils have a relatively high flash point, a selective dissolving power/attractive price. <br> - The fatty acid composition of triglycerides in vegetal oils varies due to varieties, cultivations, agronomic and climatic conditions; it is ‚green' by using smaller amounts of organic solvents or by reduced energy consumption (efficient dissolution or reduced processing times). <br> - 'Green' solvents: solvents from renewable resources/eco-friendly, water, liquid polymers, fluorinated solvents, ionic liquids/eutectic mixtures (imidazolium salts, choline acetate). |

**Table 4.** Methods for analyzing plant extracts (adapted from [42,44,45,73–76]).

| Analysis Method | Operating Conditions for Analysis | Advantages/Disadvantages |
|---|---|---|
| HPLC-high performance liquid chromatography | - It is modular in design and comprises a solvent delivery pump, auto-sampler or manual injection valve, analytical column, guard column, detector, recorder/printer;<br>- It can use a reverse-phase C18 column; UV-Vis diode array detector, and a binary solvent containing acidified water (solvent A) and a polar organic solvent (solvent B);<br>- It can include liquid-liquid partitioning with an imiscible solvent and open column chromatography, prep-HPLC and solid phase extraction. | - It is a versatile, robust, and widely used technique for the isolation of natural products; -permits a rapid processing of multicomponent samples on both analytical and preparative scale;<br>- It is preferred for separation and quantification of polyphenolics;<br>- It presents limitations especially in complex matrix, such as crude plant extracts.<br>- It is the best analytical approach to study the structure of polyphenols. |
| FTIR-Fourier transform infrared spectroscopy | - It is analyzed one drop of extract sample between two plates of sodium chloride which forms a thin film. The solvent is then evaporated off, leaving the thin film of the original material on the plate.<br>- It is used to predict the flavin, moisture content, alkaloids and phenolic substances total antioxidant capacity, catechins, free amino acids, caffeine, total polyphenols and amylose content in plant leaves (e.g., tea). | - A valuable tool for characterization and identification of compounds or functional groups (chemical bonds) present in an unknown mixture of plants extract;<br>- It is considered a powerful, fast, accurate and non-destructive tool;<br>- FTIR spectra of pure compounds are usually unique considered as a molecular, fingerprint'. |
| GC-MS | - GC analysis is carried out using a gas chromatograph equipped with FID detector, autosampler and a capillary column (DB5, 30/50 m × 0.32 mm; film thickness = 0.25 um); carrier gas ($H_2$)- 2 mL/min; programmed oven temperature = 60–200 °C, injector and detector temperatures are kept at 250° and 280 °C.<br>- GC is interfaced with an mass spectrometer (MS) with a quadruple detector, on capillary column (DB-5) (e.g., Thermoquest 2000/HP 5971, ionization voltage 70 eV) | - Qualitative analysis is based on retention times, retention indexes and mass spectra of recorded data related to computer mass spectra libraries.<br>- Quantitative data are obtained from the electronic integration of FID peak areas.<br>- GC methods for polyphenols analysis require derivatisation of VOCs.<br>- GC has a great separation capacity, offering high sensitivity and selectivity when combined with MS. |
| LC-MS/MS | - Liquid chromatography is carried out on silica gel pre-coated plates (7 mm width, F254 HPTLC) using a sample applicator at 14 mm distance from the edge of the plates; mobile phase is toluene-ethyl acetate 93:7 (v/v);<br>- Mass spectrometry (MS) is used for elucidating the chemical structures of molecules, such as peptides, polyphenols, etc.<br>- MS principle is based on ionizing chemical compounds to generate charged molecules or molecule fragments thus measuring their mass-to-charge ratios. | - LC-MS is the best analytical technique to study polyphenols in vegetal samples, and the most effective tool in study of anthocyanins structure, elucidation of procyanidin, proanthocyanidins, prodel- phinidins and tannins structure (e.g., resveratrol in wine).<br>- MS is an analytical technique relevant for structural studies on polyphenol.<br>- LC-MS/MS approach is a very powerful tool which permits anthocyanin aglycone and sugar moiety characterisation. |
| Capillary electrophoresis (CE) | - It assures the separation of neutral analytes under the influence of an electric field, and fractionation of monomeric and polymeric pigments of high molecular mass;<br>- It is used especially for determination of flavonoids in plant materials; separation of anthocyanins (a quite recently developed technique) due to their high hydrosolubility,<br>- It can be coupled with MS, being used for monitoring anthocyanins and flavonoids in wine.<br>- Micellar electrokinetic capillary chromatography (MECC) and improved gel permeation chromatography (GPC) has expended the utility to separation of neutral analytes under influence of electric field | - It is suitable for separation and qualification of low to medium molecular weight polar and charged compounds; separations are often faster and more efficient than the HPLC separations;<br>- It is a versatile analytical tool for the routine determination of a wide variety of phenolics in different types of samples;<br>- It has high separation efficiency, high resolution power; short analysis time and low consumption of sample and reagents; worse reproducibility related to chromatographic techniques; low sensitivity in terms of solute concentration; short optical path-length of capillary used as detection cell. |

**Table 4.** *Cont.*

| Analysis Method | Operating Conditions for Analysis | Advantages/Disadvantages |
|---|---|---|
| NMR spectroscopy | - Standard $^1$H, $^{13}$C and high resolution magic angle spinning (HR/MAS) NMR spectra (fingerprints) can give an overview containing a wealth of chemical information on liquid and even semi-processed extracts.<br>- Samples are prepared simply by adding 5–10% of $D_2O$ to the liquid; deuterated solvents (deuterated methanol-CD3OD, or dimethyl sulphoxide-DMSO-d6) provide a signal for magnetic field stabilisation and allow optimization of resolution of NMR peaks;<br>- Chemometric techniques are often employed to analyze data when information contained in spectra is of high complexity;<br>- NMR is very used in identifying the reaction products of anthocyanins with other compounds as cinnamic acid derivates, peroxyl radicals, catechins and flavonoids. | - It is used to foods and extracts; it has a high power of structural elucidation; it is the best non-target technique to use for screening of food extracts, mainly metabolites (fatty, amino/organic acids, sugars, aromatic compounds) detected in a single spectrum with minimal and non-distructive sample preparation.<br>- It is simple in terms of sample preparation, measurement procedure, instrumental stability, ease spectra interpretation.<br>- Selected variables (NMR peak heights or integrals) are used instead of whole spectra.<br>- The limitations of using NMR are due to the cost of equipment and low sensitivity compared with other techniques such as HPLC or GC. |
| Thermospray analysis (TSP)/Electro-spray ionisation (ESI)—MS | - TSP is proposed especially for analysing phenolic compounds, but has proven to be unsuitable for analysis of oligomers and polymers, due to thermal degradation.<br>- Direct flow injection electrospray ionisation (ESI)–MS can be used to establish polyphenol fingerprints of complex extracts. | - These advanced techniques are considered for analysing of phenolic compounds.<br>- It can not be used for analysis of high molecular weigth compounds (polymers) or procyanidic oligomers. |
| Paper chromatography (PC) | - It requires the preparation of chromatographic paper as starting line and contact points for separated extract drops, as a single drop or multiple ones, followed by immersion of the bottom of chromatography paper in specific eluate/solvent, and analysis of colored traces and quantitative estimation of results based on the calibration curves of different representative compounds. | - There are widely used for purification and isolation of anthocyanins, flavonoids, condensated tannins and phenolic acids;<br>- They are simple, quick, and inexpensive procedures;<br>- Quick answer to how many components are in a mixture (related to known compounds and spraying of phytochemical screening reagents). |
| Thin layer chromatography (TLC)/Bioautography | - It requires TLC plates with a thickness of 1 mm;<br>- It uses very little amount of sample when compared to the normal disc diffusion method and can be used for bioassay-guided isolation of compounds. | - Useful technique to determine bioactive compounds with antimicrobial activity from plant extract (direct, by contact, and in a seeded agar ovelay medium). It simplifies the process of identification and isolation. |
| Spectrophotometric methods for quantification of total phenolics | - These are based on different principles and used to determine different structural groups present in phenolic compounds;<br>- The wisely used method is the Folin-Ciocalteu assay, while the vanillin and proanthocyanidin assays are used to estimate the total proanthocyanidins. | - These methods provide very useful qualitative and quantitative data, but give only an estimation of total phenolic content;<br>- It does not separate, or give content of individual compounds;<br>- There are relative simple but are costly. |

## 5. Bio-Insecticidal Action of Some Spontaneous Flora Species in Moldova (Romania)

The region of Moldova in NE of Romania is characterized by a rich and diverse spontaneous flora, specific to subregions and growth location [35]. Environmental factors make a real mark on it, as confirmed by the particularity of growth (size, bloom, density) and the chemical composition of plants (leaves, flowers, stems or roots).

The target spontaneous flora for bio-preparations are plant species considered to be aromatic or of medicinal interest. In this context, it is interesting the information reported by Purcaru and his collaborators on the Romanian annual and perennial plants (herbaceae, trees, shrubs) [77]. They identified 64 species of native flora belonging to 21 botanical families that were studied for antimicrobial activity. Of these species, 28.1% are annual plants, 46.9% are perennial herbaceous plants and the rest (25%) are perennial woody species (shrubs and trees). Also, nearly 50% of the species belongs to the *Asteraceae* and *Lamiaceae* botanical families. From the point of view of properties that make them useful as plant protection products, antibacterial effects have been reported for 89% of species, and 57.8% of the species have shown antifungal activity [77].

Fierascu et al. performed another comprehensive review of spontaneous flora of Romania with biomedical applications [41]. Among the traditional plants, they describe the extraction methods used to obtain *Heracleum spondylium* (hogweed), *Anethum grasveolens* L. (dill), *Taraxacum officinalis* L. and *Arctium lappa* L. (burdock) extracts, in various variants, especially maceration-percolation-decoction-infusion in association with the analytical techniques for the characterization of these extracts [41]. However, in the spontaneous flora, there are elements that can also have a biopesticide effect, belonging to extremely common botanical families: *Lamiaceae*, *Asteraceae*, *Primulaceae*, *Equisetaceae*, *Hypericicaceae*, *Poligonaceae*, *Apiaceae*, etc., but the species that are parts of the Lamiaceae family numerically predominate.

From this botanical family, there are 138 species known as medicinal, aromatic or melliferous plants in Romania. The most common of these are the mint, nettle, basil, sage, and thyme species, which have been sometimes used in popular agricultural practices to combat pests [46,77–80].

It is an increasing number of literature studies that highlight the bio-pesticide potential of plant extracts obtained from spontaneous or cultivated flora in Romania (Table 5). Numerous experimental studies have been reported on the composition of plant extracts from the spontaneous flora of Romania, as well as practice reports on the use of these plants extracts to combat the crops pest or grain stores and seeds. Thus, Morar et al. studied the effect of nicotine hydroalcoholic extracts (from *Nicotiana tabacum* and *Nicotiana rustica*), anabasina (from *Anabasis aphylla*), quasina (from *Quasina amara*), rotenone (from *Derris elliptica*), and pyrethrin (*Chrysanthenum cinerariaefolium*, *Pyretrum coccine*) on Colorado beetle (*Leptinotarsa decemlineata* Say.) [80].

Brudea et al. studied the efficacy of secondary metabolites extracted from native plants of the common fern (*Driopteris filix mas*), sage (*Salvia nemorosa*), wormwood (*Artemisia dracunculus*, *A. vulgaris*, *A. absinthium*), the wolf (*Aristolochia clematidis*), *Heracleum spondylium*, *Stachis sylvatica*, *Tanacetum vulgare*, *Urtica dioica*, *Sambucus ebulus*, and *Taxus baccata* on caterpillar (*Hyphantria cunea Drury)* (*F. Arctiidae-Lepidoptera*) under laboratory conditions [81].

Daraban et al. studied the potential repellent and/or germicidal effect of spontaneous flora extracts (Satureja hortensis, Ocimum basilicum, Origanum vulgare, Hypericum perforatum, Rumex patientia, Achillea millefolium, Calendula officinalis, Matricaria chamomilla, Salvia officinalis, Pimpinella anisum, Equisetum arvense, Allium sativum, Urtica dioica, Primula veris) on pests (Leptinotarsa decemlineata) and insects from deposits (Acanthoscelides obsoletus) [81,82].

Coisin et al. make a chemical and phytochemical assessment of some sage species from Romania [79], and Pruteanu and his collaborators performed a biochemical and physicochemical comparative study of four types of extracts obtained from dry air thyme (*Thymus serpyllum L harvested*) of spontaneous Romanian flora [83].

**Table 5.** Practical use of extracts from Romanian spontaneous flora.

| Type of Extract | Effect on Targets | Observations | Ref. |
|---|---|---|---|
| ● Hydroalcoholic extracts (20%) of:<br>- *Chrysanthemum cinerariaefolium* Trev.<br>- *Chrysanthemum balsamita*,<br>- *Ruta corsica*,<br>- *Artemisia absinthium* L.,<br>- *Taracxacum officinale* L.,<br>- *Tagetes erecta* L. | Colorado beetle (*Leptinotarsa decemlineata* Say.) specific to potato crops Rejective effect for adults who does not attack the crops | Efficiency against larvae: 99.01%, 93.6% and 96.83% Efficiency of 7.14% (poor) till 72.22% (good) | [31,80] |
| ● Aqueous and alcoholic extracts of:<br>- *Driopteris filix* mas (fern),<br>- *Salvia nemorosa*,<br>- *Artemisia dracunculus*,<br>- *A. vulgaris*,<br>- *A. absinthium*,<br>- *Aristolochia clematidis*,<br>- *Heracleum spondylium*,<br>- *Stachis sylvatica*,<br>- *Tanacetum vulgare*,<br>- *Urtica dioica*,<br>- *Sambucus ebulus* and<br>- *Taxus baccata*. | Webworm *Hyphantria cunea* Drury (*F. Arctiidae–Lepidoptera*). Results: Extracts in ethanol (*Sambucus ebulus*, *Artemisia vulgaris*, *A. absinthium*, *Tanacetum vulgare*, *Urtica dioica*, *Aristolochia clematidis*, *Heracleum spondylum*, *Taxus baccata*, *Salvia nemorosa*) reduced the larvae production; extracts in cold water reduced webworm adults activity (*Artemisia vulgaris*, *A. Dracunculus*, *Salvia nemorosa*, *Stachys silvatica*). | The extracts were obtained from crushed/milled plants, using 25 g/L of cold water, and under stirring for 24 h. Ethyl alcohol extracts were obtained from 25 g of dry plants/200 mL of alcohol, filled to 1 L with water. The experiments were carried out under laboratory conditions (application on caterpil-lock shoots, placed in growth jar test vessels) | [78] |
| ● Alcoholic macerates of:<br>- *Salvia officinalis*,<br>- *Ocimumbasilicum, Satureja hortensis*,<br>- *Origanum vulgare*,<br>- *Primula veris*,<br>- *Equisetum arvense*,<br>- *Urtica dioica*,<br>- *Allium sativum*,<br>- *Pimpinella anisum*,<br>- *Matricaria chamomilla*,<br>- *Calendula officinalis*,<br>- *Achillea millefolium*,<br>- *Hypericum perforatum*,<br>- *Rumex patientia*,<br>- *Achillea millefolium*, | *Leptinotarsa decemlineata* say and their larvae at various stages of development.<br>Laboratory results: the most effective extract was that of *Primula veris*, which demonstrated a 100% killing rate for both adults and larvae in the first 24 h in a 5 L-location space. *Hypericum perforatum*, *Achilleia milefolium* and *Pimpinella anisum* have shown low efficiency for both adult Colorado beetles and their larvae, but these can also reduce the number of *Leptinotarsa decemlineatasay* in organic crop. | Plants were firstly ground/crushed, than screened and weighted with an analytical balance: 10 g of each plant sample. After these were treated with 100 mL of 97% ethylic alcohol in an Erlenmeyer vessel. The mixtures were periodically stirred and the maceration time was of minimum 90 days. The number of larvaes and adults was periodically counted for determination of its biopesticidal efficiency against larvae and adults. Different extract dispersion ways were tested for insecticidal effect | [81] |
| - *Hypericum perforatum* and<br>- *Origanum vulgare* | Insect beans-*Acanthoscelides obsoletus* | | [82] |

## 6. Conclusions

Although plant protection chemicals (pesticides) were initially used to increase agricultural productivity, to control pests and infectious diseases, but also to eliminate parasitic vegetation, their negative effects have been shown to be cumulative over time, meaning increased risks vis-à-vis the health of living beings, but also a series of negative effects on the environment, thus counterbalancing the benefits associated with their use. Some of the side effects associated with the application of plant protection chemicals have occurred in the form of increasing the population of resistant pests, declining beneficial organisms such as predators, pollinators, and earthworms, changes in soil microbial diversity, contamination of the water and air ecosystem, and last but not least, affecting human health by reducing the quality of food resulting from the processing of agricultural raw materials.

In this review, we intended to paint an exhaustive picture of the information in the literature on methods of obtaining plant extracts, specifically concerning chemical and physico-chemical methods for their qualitative and quantitative characterization by identifying those biologically active compounds that determine the pesticide action of these types of extracts.

The results of this review highlight the fact that there is a growing challenge in the development and characterization of natural products for the protection of plant crops and seeds in deposits (biopesticides), which will largely involve bio-nanotechnology in the full study of resistant genotypes or pesticides with fewer adverse effects, as well as sustainable and long-term agricultural practices. It is also particularly useful the information that facilitating the growth of spontaneous flora will be able to lead in the future to high-performance products with reduced costs, an essential element that will facilitate the use of such products by small farmers.

**Author Contributions:** D.S.—review coordinator and Sections 1 and 5; C.Z.—Abstract, Sections 3 and 4, the manuscript translation in English and the entire correction of the manuscript; L.R.—Section 2, Conclusions and the manuscript translation in English; M.B.—responsible and consultant for the part related to plants and plant protection; G.M.D.—ensured the collection, selection and formatting of data from the literature. All authors have read and agreed to the published version of the manuscript.

**Funding:** This research received no external funding.

**Conflicts of Interest:** The authors declare no conflict of interest.

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
