# Peer review of "Challenge of Utilization Vegetal Extracts as Natural Plant Protection Products"

_applsci, doi:10.3390/app10248913_

Round 1

Reviewer 1 Report

Line 17: biopesticides - can be replaced with the more correct wording 'natural plant protection products' (this applies to the entire work) Line 18: what biopesticides are made from animal animals? Is it legal? Line 19: chemical pesticides - unfortunate formulation, as it is obvious that pesticides are chemical, therefore you can use: 'pesticides' or 'chemical plant protection products' Line 66/67: not only extracts with chrysanthemums, it is worth expanding this aspect, then the work will gain more value. It is worth mentioning the extracts of phenolic compounds obtained from various plants - the literature on this topic is extensive and available on the Internet Line 73: 'The paper aim is to synthesize pieces of information' - change to 'the aim of the work is to systematize information' Line 80: 'the herbal extracts from the spontaneous flora of Moldova' - what does it mean for spontaneous flora? Line 85: '..are used in agriculture to reduce, destroy and / or kill a number of crops or deposit pests, such as insects, fungi, weeds or another unwanted micro / macro organisms ..' - please sort it out, e.g. '… Are used in agriculture to prevent the growth and destruction of weeds and pests that affect crop quality. Among the factors determining the quality and quantity of crops are, among others: wild animals, rodents, insects and, most of all, pathogenic microorganisms, such as: fungi (Fusarium spp. Aspergillus spp.), Bacteria (add a few examples) and viruses. [search for literature] Line 87: 'The action of pests, of any category, is to damage crops, disturb the balance of crops and agricultural production and, affect the human health by negative influence on the quality of agrifood products' -> The action of harmful factors is primarily on the reduction of the quality and quantity of crops and obtained food products, which may affect the health of the consumer. Line 155: 'natural organic pesticides' -> natural plant protection products Line 190-196: this can be put in the table, it will be easier to read Line 216: literature [1,15,20,21,29,32] - I suggest assigning individual literature items to the appropriate paragraph (in lines 206-216) Table 1 and Table 2: the methodology can be more generalized and the literature in which it is described in detail is cited, i.e. a review, so there is no need to provide details of the extraction as they were not developed by the authors of this paper. These tables will then be shorter and the work will become clearer Figure 2: I propose to give up the colors and make a black and white drawing, or to give up shading and use bright colors - it will be more legible Line 419: 'Fierascu and his collegues' - this is not a professional statement, you can use 'Fierascu et al.' Table 4: It's okay, even more literature can be added - it's available To sum up: The manuscript is written correctly and to the topic, but in order to be accepted, please take into account the comments and supplement the available literature on the subject, then this work will be of greater value for science.

Author Response

Dear Sir/Madam,

Thank you for reviewing our manuscript and for all your comments and suggestions that have been helpful to improve our paper quality. All new changes made in the manuscript were highlighted in red.

Reviewer Comments (R1 Ci) / Authors answer (A Ai)

R1 C1 Line 17: biopesticides - can be replaced with the more correct wording 'natural plant protection products' (this applies to the entire work)

A A1 The term of ‘biopesticides’ was replaced with the new terminology of ‘natural plant protection products’ at line 17, and also, when possible, in the whole manuscript text.

R1 C2 Line 18: what biopesticides are made from animal? Is it legal?

A A2 Biopesticides derived from certain insects, worms, parts of animal organs, etc. (e.g., microscopic roundworms (Entomopathogenic nematodes), spinosad (Saccharopolyspora spinosa), insect hormones and insect growth regulators). It is legal caused are valorized the useful constituents from organs of died animals or cropped insects and worms in special locations without affecting the safe and quality of environment.

R1 C3 Line 19: chemical pesticides - unfortunate formulation, as it is obvious that pesticides are chemical, therefore you can use: 'pesticides' or 'chemical plant protection products' .

A A3 The term of ‘chemical pesticides’ was replaced with the term of ‘chemical plant protection products’ at line 19, and also, when possible, in the whole manuscript text.

R1 C4 Line 66/67: not only extracts with chrysanthemums, it is worth expanding this aspect, then the work will gain more value. It is worth mentioning the extracts of phenolic compounds obtained from various plants - the literature on this topic is extensive and available on the Internet

A A4 It was completed the large field of extracts with others (see line 66/67) including extracts of phenolic compounds from various plants.

R1 C5 Line 73: 'The paper aim is to synthesize pieces of information' - change to 'the aim of the work is to systematize information'

A A5 It was changed the text and replaced with suggested formulation at line 73.

R1 C6 Line 80: 'the herbal extracts from the spontaneous flora of Moldova' - what does it mean for spontaneous flora?

A A6 It was called "spontaneous flora" those plants that are not cultivated by humans, do not grow in a controlled system but grow freely, wild in nature in different regions.

R1 C7 Line 85: '…are used in agriculture to reduce, destroy and / or kill a number of crops or deposit pests, such as insects, fungi, weeds or another unwanted micro / macro organisms ..' - please sort it out, e.g. '… Are used in agriculture to prevent the growth and destruction of weeds and pests that affect crop quality. Among the factors determining the quality and quantity of crops are, among others: wild animals, rodents, insects and, most of all, pathogenic microorganisms, such as: fungi (Fusarium spp. Aspergillus spp.), Bacteria (add a few examples) and viruses. [search for literature]

A A7 The suggested text was inserted in the manuscript text, as can see at lines 93-97 and some examples of bacteria and viruses were inserted.

R1 C8 Line 87: 'The action of pests, of any category, is to damage crops, disturb the balance of crops and agricultural production and, affect the human health by negative influence on the quality of agrifood products' -> The action of harmful factors is primarily on the reduction of the quality and quantity of crops and obtained food products, which may affect the health of the consumer.

A A8 The text was changed as ‘The action of pests, of any category, is primarily to reduce the quality and quantity of crops and obtained food products, which may affect the health of consumer (e.g., human health)’.

R1 C9 Line 155: 'natural organic pesticides' -> natural plant protection products

A A9 The term of 'natural organic pesticides' was changed with that of ‘natural plant protection products’ at line 169.

R1 C10 Line 190-196: this can be put in the table, it will be easier to read

A A10 The sentence was modified and a part of data were inserted in a table (table 1) at line 211

R1 C11 Line 216: literature [1,15,20,21,29,32] - I suggest assigning individual literature items to the appropriate paragraph (in lines 206-216)

Table 1 and Table 2: the methodology can be more generalized and the literature in which it is described in detail is cited, i.e. a review, so there is no need to provide details of the extraction as they were not developed by the authors of this paper. These tables will then be shorter and the work will become clearer

Figure 2: I propose to give up the colors and make a black and white drawing, or to give up shading and use bright colors - it will be more legible

A A11 The literature was inserted individually in corresponding paragraph (lines 206-216).

In tables 1 and 2, the text was simplified.

The colors in Figure 2 were clarified, only shading from two colours used (white and black).

R1 C12 Line 419: 'Fierascu and his collegues' - this is not a professional statement, you can use 'Fierascu et al.' Table 4: It's okay, even more literature can be added - it's available

A A12 We changed the text considering your comment at line 419 and also in other sentences were presented.

R1 C13 To sum up: The manuscript is written correctly and to the topic, but in order to be accepted, please take into account the comments and supplement the available literature on the subject, then this work will be of greater value for science.

A A12 Thank you for your encouraging remarks. We considered your comments and suggestions for improving our manuscript quality.

Once again, we are thankful for your feedback and suggestions that have been useful to provide a better version of the manuscript.

Reviewer 2 Report

This topic is of interest to Applied Sciences readers. However, changes are needed before this should be considered for publication.

The abstract should include more content concerning some background for the research, but first of all it lacks the objective of the research. Please indicate the main conclusions or interpretations. Generally speaking, the abstract's content is insufficient.

The title of the article does not reflect the content of the article. The authors do not explain the concept of sustainable agriculture, what are the main assumptions and challenges (as the title suggests). The paper focuses primarily on presenting information about pesticides, biopesticides, analysis of the main strengths and weaknesses that result from the use of plant extracts as biopesticides, methods used to obtain plant extracts etc. I suggest making a correction to the title of the manuscript or the content of the article to refer to the concept of sustainable agriculture and indicate what the challenges are.

The chapter numbering is wrong

Figure 2. Do the colors used in the drawing have any meaning? If so, the legend is missing.

I don't know what the fragments marked in yellow mean in the text.

Your conclusion section reads more like an abstract.

Author Response

List of responses for Reviewer R2 (1st revised manuscript)

Dear Sir/Madam,

Thank you for reviewing our manuscript and for all your comments and suggestions that have been helpful to improve our paper quality. All new changes made in the manuscript were highlighted in red.

Reviewer Comments (R2 Ci) / Authors answer (A Ai)

R2 C1 The abstract should include more content concerning some background for the research, but first of all it lacks the objective of the research. Please indicate the main conclusions or interpretations. Generally speaking, the abstract's content is insufficient.

A A1 The abstract content was improved and the research objectives were mentioned.

R2 C2 The title of the article does not reflect the content of the article. The authors do not explain the concept of sustainable agriculture, what are the main assumptions and challenges (as the title suggests). The paper focuses primarily on presenting information about pesticides, biopesticides, analysis of the main strengths and weaknesses that result from the use of plant extracts as biopesticides, methods used to obtain plant extracts etc. I suggest making a correction to the title of the manuscript or the content of the article to refer to the concept of sustainable agriculture and indicate what the challenges are.

A A2 You have right. The title must be modified accordingly. We will ask the editors to agree and accept the change of our manuscript title as Challenge of utilization vegetal extracts as natural plant protection products

R2 C3 The chapter numbering is wrong

A A3 We correct the chapters numbering in our manuscript.

R2 C4 Figure 2. Do the colors used in the drawing have any meaning? If so, the legend is missing.

A A4 We changed the colors in Figure 2, it rested only black and white colors and derived colors with shadows.

R2 C5 I don't know what the fragments marked in yellow mean in the text.

A A5 Sorry for our mistake. The text and references were again verified for correctness.

R2 C6 Your conclusion section reads more like an abstract.

A A6 We tried to complete the conclusions section and hope it will be satisfactory for you.

Once again, we are thankful for your feedback and suggestions that have been useful to provide a better version of the manuscript.

Reviewer 3 Report

Manuscript ID: insects-960555

Title: A challenge for future sustainable agriculture and quality food products: vegetal extracts

Suteu et al. present a review about plant extract biopesticides, authors have tried to summarize and highlight the properties and applications, advantages/ in relation to synthetic pesticides, action mechanism, physical-chemical analysis methodologies, characteristics, advantages/disadvantages. This manuscript can be a useful paper for people that work or will start to work in biopesticides, related to plant extract biopesticides and physical-chemical analysis methodologies used.

In general, the manuscript is well written, but I miss some recent literature in the manuscript. I suggest revising the literature related to this paper published in 2020…for instance: Yadav et al. 2020. Plant microbiomes for sustainable agriculture. https://doi.org/10.1007/978-3-030-38453-1  and Ke et al. 2020. Biology of plant-associated microbiomes in sustainable agriculture. https://doi.org/10.1016/j.tibtech.2020.07.008.

I found some parts repetitive, when I was reading, I got many time the idea that I already read it before; in my opinion, tables also should be improved/summarized; uniformization, as for instance use or not the author after scientific names, criterion to use the italic letter; many parts with double space. In my opinion, the conclusion section should be rethought and improved, the present version seems a short summary of introduction…nothing new, for my not a conclusion from a literature revision at all!

Below are some minor comments:

L34-L35: For me, the first sentence in the introduction makes no sense.

L39: …respectively?

L46: instead of insect use arthropods

L86: …kill a number of crops?  maybe pests

L88-89: The action of pests, of any category, is to damage crops, disturb the balance of crops and agricultural production. Maybe better: The action of pests, of any category, is to damage, disturb the balance and production of crops.

L119: fulfill

L166: Phytophthoras, remove s

L168: natural mushrooms change by entomopathogenic fungus

L168: remove italic in Entomopathogenic nematodes

L171: instead of removal use control

L181: remove italic in phytoalexins and in L183 in phytoanticipins

L182: De instead of in use de novo; and it should be in italic

L183: The authors?  Pino et al 2013?

L190: nicotine, remove the italic here and then

L195-196: …or for fumigant toxicity of C. chinensis essential oil (EO)… I did not get it!

L195: C. chinensis, Callosobruchus chinensis?

L196-197: S. Tenuifolia …. L. Ingenue;  Senegalia tenuifolia? Stephanomeria tenuifolia?… Lycoriella ingenue

L200: (Miller) be consistent if use author here…please use also in the previous scientific names 

L208: Matran, ® registered trademark?

L215: GreenMatch EXTM, ® registered trademark?

L222-L223: …to be destroyed by shock effects,… I did not get it?

L278: Althought, remove t

L328: Table 1: Extraction / Analysis method; In this section, in my opinion, should be more summarized.

L356: repetable, should be repeatable

Table 2: -        In Characteristics / Advantages / Disadvantages.  It is expensive due to the need to dispose off large amounts of organic waste (waste solvent), which in itself can risk environmental issues. I did not get it! Please revise English!

Table 2: in SCFE-based method: Characteristics / Advantages / Disadvantages; recircu-lated, remove hyphen

L374: Figure 2 should be improved

L390: Table 1?,  I did not find it in table 1

L405: Table 3: Summarize, and I have some doubt about some advantages and disadvantages presented

Table 3: imiscible in Operating conditions imposed for analysis should be immiscible

Table 3: characteriza-tion, remove hyphen

L413: herbaceae, should be herbaceous

L420: collegues, should be colleagues

L422: Heracleum spondylium (author?)

L428: …just used italic in families, Lamiaceae ?

L435: combate, remove e

L444: F. Arctiidae – Lepidoptera;  family?

L454: A bit confusing, as the other tables used and not summarise much!

L455: As I have mentioned before, the conclusion section should be improved, seems a short summary of the introduction…nothing new, for my not a conclusion from a literature revision at all!

L457: …parasitic vegetation?; weeds?

L460: resistant pests?; pest resistance?

Author Response

List of responses for Reviewer R3 (1st revised manuscript)

Dear Sir/Madam,

Thank you for reviewing our manuscript and for all your comments and suggestions that have been helpful to improve our paper quality. All new changes made in the manuscript were highlighted in red.

Reviewer Comments (R3 Ci) / Authors answer (A Ai)

R3 C1 In general, the manuscript is well written, but I miss some recent literature in the manuscript. I suggest revising the literature related to this paper published in 2020…for instance: Yadav et al. 2020. Plant microbiomes for sustainable agriculture. https://doi.org/10.1007/978-3-030-38453-1  and Ke et al. 2020. Biology of plant-associated microbiomes in sustainable agriculture. https://doi.org/10.1016/j.tibtech.2020.07.008.

A A1 The recommended two articles were inserted as references for this manuscript [69-70].

R3 C2 I found some parts repetitive, when I was reading, I got many time the idea that I already read it before; in my opinion, tables also should be improved/summarized; uniformization, as for instance use or not the author after scientific names, criterion to use the italic letter; many parts with double space. In my opinion, the conclusion section should be rethought and improved, the present version seems a short summary of introduction…nothing new, for my not a conclusion from a literature revision at all!

A A2 The quality of manuscript was improved. The tables were shortened and information presented in the same manner – uniformity of data presentation was done; a new table was inserted as recommendation of reviewer #1 thus the tables’ number has been changed. We hope that the conclusions part will be satisfactory for you in the present format.

R3 C3 L34-L35: For me, the first sentence in the introduction makes no sense.

A A3 Sorry for our mistakes. The first sentence was deleted.

R3 C4 L39: …respectively?

A A4 It was deleted the word ‘respectively in the sentence line 39.

R3 C5 L46: instead of insect use arthropods

A A5 It was changed the term of insect with arthropods at line 46.

R3 C6 L86: …kill a number of crops?  maybe pests

A A6 The text was modified accordingly.

R3 C7 L88-89: The action of pests, of any category, is to damage crops, disturb the balance of crops and agricultural production. Maybe better: The action of pests, of any category, is to damage, disturb the balance and production of crops.

A A7 The manuscript text at L88-89 was modified as ‘The action of pests, of any category, is primarily to reduce the quality and quantity of crops and obtained food products, which may affect the health of consumer (e.g., human health)’.

R3 C8 L119: fulfill – changed in text;

L166: Phytophthoras, remove s – change done;

L168: natural mushrooms change by entomopathogenic fungus – the text was changed ;

L168: remove italic in Entomopathogenic nematodes – it was removed italic ;

L171: instead of removal use control – was used control instead of removal;

L181: remove italic in phytoalexins and in L183 in phytoanticipins – it was removed italic.;

L182: De instead of in use de novo and it should be in italic  – it was changed the word and it is written in italic; L83: The authors?  Pino et al 2013? – Yes, this is the reference.

L195-196: …or for fumigant toxicity of C. chinensis essential oil (EO)… I did not get it! L195: C. chinensis, Callosobruchus chinensis? - Yes

L196-197: S. Tenuifolia …. L. Ingenue;  Senegalia tenuifolia? Yes Stephanomeria tenuifolia?… Lycoriella ingénue Yes

 L200: (Miller) be consistent if use author here…please use also in the previous scientific names – it was mention the researcher name.

L208: Matran, ® registered trademark? -  Matran EC

L215: GreenMatch EXTM, ® registered trademark? - GreenMatch EXTM

L222-L223: …to be destroyed by shock effects,… I did not get it? – Text modified

L278: Althought, remove t – correction was done.

A A8 The recommended corrections were done in the manuscript text.

R3 C9 L328: Table 1: Extraction / Analysis method; In this section, in my opinion, should be more summarized.

A A9 The table 1 was summarized .

R1 C10 Table 2: - In Characteristics / Advantages / Disadvantages.  It is expensive due to the need to dispose off large amounts of organic waste (waste solvent), which in itself can risk environmental issues. I did not get it! Please revise English!

Table 2: in SCFE-based method: Characteristics / Advantages / Disadvantages; recircu-lated, remove hyphen

A A10 The English was revised in Table 2 and removde hyphen.

R3 C11 L374: Figure 2 should be improved

A A11 The Figure 2 was improved – only white and black colors used.

R3 C12 L390: Table 1?,  I did not find it in table 1 – It was corrected the position in text.

L405: Table 3: Summarize, and I have some doubt about some advantages and disadvantages presented

Table 3: imiscible in Operating conditions imposed for analysis should be immiscible – it was corrected

Table 3: characteriza-tion, remove hyphen – Yes, it was corrected.

A A12 We summarized the text in table 3. All corrections were done in the manuscript text.

R3 C13 L413: herbaceous, should be herbaceous

L420: collegues, should be colleagues – we corrected in text

L422: Heracleum spondylium (author?) – we inserted the author.

L428: …just used italic in families, Lamiaceae ? – Yes, it was corrected.

L435: combate, remove e

L444: F. Arctiidae – Lepidoptera;  family? - Yes

L454: A bit confusing, as the other tables used and not summarise much! – The tables were simplified in order to be not confusing.

L457: …parasitic vegetation?; weeds?

L460: resistant pests?; pest resistance?

A A12 Thank you for your recommendations. We corrected them in the manuscript text.

Once again, we are thankful for your feedback and suggestions that have been useful to provide a better version of the manuscript.

Reviewer 4 Report

This is an interesting review paper about the potential use of non-chemical ingredients for controlling plant pests. The ms has suitable structure, and the English is suitable. After careful reading of the ms I found it suitable for publication, however before acceptation the text need to be improved in the followoning parts:

p. 5, line 166: "spp." need to be written with normal letters, not in italic

p. 5, line 169: I suggest to add a reference about entomopathogenic nematodes, for example: LAZNIK et al., 2010. Control of the Colorado potato beetle (Leptinotarsa decemlineata [Say]) on potato under field conditions: a comparison of the efficacy of foliar application of two strains of Steinernema feltiae (Filipjev) and spraying with thiametoxam. Journal of plant diseases and protection, 117, 3: 129-135.

p. 5, line 197: I suggest to add a reference or two about active ingredients of biopesticides, for example: ROJHT et al., 2012. Chemical analysis of three herbal extracts and observation of their activity against adults of Acanthoscelides obtectus and Leptinotarsa decemlineata using a video tracking system. Journal of plant diseases and protection, 119, 2: 59-67.

p. 6, line 219: I suggest to add some references about biopesticides, which have been insufficiently studies up to now, for example: BOHINC et al., 2020. The first evidence of the insecticidal potential of plant powders from invasive alien plants against rice weevil under laboratory conditions. Applied sciences, 10, 21, art. 7828.

Author Response

List of responses for Reviewer R4 (1st revised manuscript)

Dear Sir/Madam,

Thank you for reviewing our manuscript and for all your comments and suggestions that have been helpful to improve our paper quality. All new changes made in the manuscript were highlighted in red.

Reviewer Comments (R4 Ci) / Authors answer (A Ai)

R4 C1 p. 5, line 166: "spp." need to be written with normal letters, not in italic

A A1 Thank you for observation. The correction was done in the manuscript.

R4 C2 p. 5, line 169: I suggest to add a reference about entomopathogenic nematodes, for example: LAZNIK et al., 2010. Control of the Colorado potato beetle (Leptinotarsa decemlineata [Say]) on potato under field conditions: a comparison of the efficacy of foliar application of two strains of Steinernema feltiae (Filipjev) and spraying with thiametoxam. Journal of plant diseases and protection, 117, 3: 129-135.

A A2 The recommended reference was inserted in the manuscript.

R4 C3 p. 5, line 197: I suggest to add a reference or two about active ingredients of biopesticides, for example: ROJHT et al., 2012. Chemical analysis of three herbal extracts and observation of their activity against adults of Acanthoscelides obtectus and Leptinotarsa decemlineata using a video tracking system. Journal of plant diseases and protection, 119, 2: 59-67.

A A3 Thank you for your recommendation. It was not added as reference due to the high number of existing references in the manuscript.

R4 C4 p. 6, line 219: I suggest to add some references about biopesticides, which have been insufficiently studies up to now, for example: BOHINC et al., 2020. The first evidence of the insecticidal potential of plant powders from invasive alien plants against rice weevil under laboratory conditions. Applied sciences, 10, 21, art. 7828.

A A4 Thank you for your recommendation. It was not added as reference due to the high number of existing references in the manuscript.

Once again, we are thankful for your feedback and suggestions that have been useful to provide a better version of the manuscript.

Round 2

Reviewer 2 Report

All comments contained in my review have been taken into account by the Authors. Thank you